Whole-brain ex-vivo quantitative MRI of the cuprizone mouse model

http://orcid.org/0000-0001-7640-5520 Wood Tobias C. 1 tobias.wood@kcl.ac.uk
http://orcid.org/0000-0002-8561-4241 Simmons Camilla 1
Hurley Samuel A. 2 3
Vernon Anthony C. 4
Torres Joel 1
Dell’Acqua Flavio 1 5
Williams Steve C.R. 1
Cash Diana 1
1 Department of Neuroimaging, IOPPN, King’s College London , London , United Kingdom
2 FMRIB Centre, Nuffield Department of Clinical Neurosciences, University of Oxford , Oxford, Oxfordshire , United Kingdom
3 Synaptive Medical , Toronto, ON , Canada
4 Cells and Behaviour Unit, Department of Basic and Clinical Neuroscience, IOPPN, King’s College London , London , United Kingdom
5 NatBrainLab, Department of Basic and Clinical Neuroscience, IOPPN, King’s College London , London , United Kingdom
Abdullah Jafri
Electronic publication date: 2016 Nov 1
Publication date: 2016
Volume: 4
Electronic Location ID: e2632
Received 2016 Jul 28; Accepted 2016 Sep 29
Copyright: © 2016 Wood et al.
Copyright year: 2016
Copyright holder: Wood et al.
License: This is an open access article distributed under the terms of the Creative Commons Attribution License, which permits unrestricted use, distribution, reproduction and adaptation in any medium and for any purpose provided that it is properly attributed. For attribution, the original author(s), title, publication source (PeerJ) and either DOI or URL of the article must be cited.
License URL: https://creativecommons.org/licenses/by/4.0/

Keywords: Cuprizone, Myelin, Quantitative imaging, MRI, Inflammation, Mouse

Funding: MRC G0800298 The study received funding from the MRC (grant code G0800298). The funders had no role in study design, data collection and analysis, decision to publish, or preparation of the manuscript.

==============================
Myelin is a critical component of the nervous system and a major contributor to contrast in Magnetic Resonance (MR) images. However, the precise contribution of myelination to multiple MR modalities is still under debate. The cuprizone mouse is a well-established model of demyelination that has been used in several MR studies, but these have often imaged only a single slice and analysed a small region of interest in the corpus callosum. We imaged and analyzed the whole brain of the cuprizone mouse ex-vivo using high-resolution quantitative MR methods (multi-component relaxometry, Diffusion Tensor Imaging (DTI) and morphometry) and found changes in multiple regions, including the corpus callosum, cerebellum, thalamus and hippocampus. The presence of inflammation, confirmed with histology, presents difficulties in isolating the sensitivity and specificity of these MR methods to demyelination using this model.

Introduction

Myelin is a critical component of a healthy nervous system. It is composed of protein and lipid layers that tightly wrap neurons, improving their electrical conductivity and reducing their energy requirements (Nave & Werner, 2014). Focal demyelinating lesions are the hallmark of Multiple Sclerosis and disruption of myelin is also associated with other neurodegenerative diseases such as Alzheimer’s and Parkinson’s Disease. Non-invasive methods to quantify the myelination state of the nervous system are hence highly useful in order to better track the progression of these diseases, and any protective or regenerative treatments that become available (Dubessy et al., 2014).

Myelin is also a uniquely useful structure for MRI as it contributes to almost every known contrast mechanism. The high lipid content provides abundant pathways for spin-lattice interactions, reducing the longitudinal relaxation time (T1) of surrounding protons (Stuber et al., 2014) and generating significant Magnetization Transfer (MT) effects (Turati et al., 2014). It is impermeable, hindering the diffusion of water molecules (Song et al., 2005). Myelin is diamagnetic compared to grey matter (GM), leading to excellent phase and susceptibility contrast (Lee et al., 2012). The unique layered structure traps water protons in restricted environments, leading to reduced transverse relaxation (T2) and an additional inhomogeneous or dipolar MT effect (Mackay et al., 1994; Varma et al., 2015). It is hence of little surprise to find great interest in quantifying myelin content using Magnetic Resonance (MR) techniques.

Although numerous approaches have been proposed to quantify myelin with MRI, proper validation of such methods can be lacking in the literature. A particular case in point is the multi-component Driven-Equilibrium Single-Pulse Observation of T1/T2 (mcDESPOT) method. This aims to divide the MR signal resulting from steady-state sequences into three pools representing water trapped in the myelin sheath, intra-extracellular water, and free water in cerebrospinal fluid (Deoni, Matthews & Kolind, 2013). It has been used in several clinical studies, however to our knowledge it has had little validation with pre-clinical or histological studies. It has also been criticized for theoretical difficulties with the fitting procedure (Lankford & Does, 2012; Zhang et al., 2014), but these have been partially addressed by subsequent literature (Hurley & Alexander, 2014; Bouhrara & Spencer, 2015). In this paper, we provide a practical demonstration of mcDESPOT in the widely-used cuprizone mouse model of demyelination (Torkildsen et al., 2008; Skripuletz et al., 2011). In this model mice are fed cuprizone over a period of weeks, and progressive demyelination can be observed throughout white matter (WM) (Goldberg et al., 2015).

The cuprizone model has already been extensively imaged with multiple MR techniques, which are summarized in Table 1. However, the majority of these studies were restricted to imaging just one or two slices of the brain, often chosen to cover sections of the corpus callosum. This negates one of the major benefits of MRI, which is the ability to image the entire brain in a reasonable amount of time. The analyses are then often conducted on a small number of Regions-Of-Interest (ROIs), reducing the rich information available in MRI to a single composite number.

Table 1 A review of the existing cuprizone literature.

A wide variety of MR modalities have been used, but a majority of papers analyzed a small ROI in the corpus callosum.

Citation	In-/Ex-vivo	Acquired volume	Analysis volume	Methods	
Song et al. (2005)	Ex	MS	CC, EC, OT, CP	DTI	
Merkler et al. (2005)	In	3D	CC, EC	T1w, T2w, MTR	
Sun et al. (2006)	In	MS	CC	DTI	
Wu et al. (2008)	In	MS	CC	T2w, DTI	
Torkildsen et al. (2009)	In	MS	Lesion volumes	T2w	
Xie et al. (2010)	In	MS	CC	DTI	
Zhang et al. (2012)	Both	3D/MS	CC, Ctx	T2w, MTR, DTI	
Chandran et al. (2012)	In	MS	CC, Cg, EC	T2w, DTI	
Thiessen et al. (2013)	Both	SS	CC, EC, Ctx	T1/2, qMT, DTI, MWF	
Fjær et al. (2013)	In	3D	CC, GM, Cbl, Ctx	T2w, MTR	
Falangola et al. (2014)	In	MS	CC	DKI	
Turati et al. (2014)	In	SS×2	CC, EC	qMT	
Guglielmetti et al. (2016)	In	MS	CC, Ctx	T2w, DKI	
Jelescu et al. (2016)	In	SS	CC	T2w, MTR, DKI	
Tagge et al. (2016)	In	3D	Cbr	T2w, MTR	
Note:

SS, Single-slice; MS, Multi-slice; CC, Corpus Callosum; EC, External Capsule; Ctx, Cortex; Cg, Cingulum; Cbl, Cerebellum; Cbr, Cerebrum; OT, Optic Tracts; CP, Cerebral Peduncles; T1w, T1 Weighted; T2w, T2 Weighted; T1, T1 Map; T2, T2 Map; MTR, Magnetisation Transfer Ratio; qMT, Quantitative Magnetisation Transfer; DTI, Diffusion Tensor Imaging; DKI, Diffusion Kurtosis Imaging; MWF, Myelin Water Fraction.

Hence a further aim of this study was to acquire full-brain, high resolution MRI of the cuprizone model, and look for effects outside of the corpus callosum. Such methodology is widespread in clinical MR studies, so their use in pre-clinical studies will aid translatability. We are aware of only two cuprizone papers that attempted similar acquisitions; however, Tagge et al. (2016) did not achieve sufficient SNR in the cerebellum and Fjær et al. (2013) did not conduct histology there.

The comprehensive study of Thiessen et al. (2013) attempted to measure the Myelin Water Fraction (MWF) with the quantitative Multi-Echo T2 (MET2) method but failed to find a MWF even in healthy WM at 7T. MET2 is considered somewhat of a gold standard for MWF imaging, so this study was an opportunity to evaluate whether mcDESPOT could robustly detect a MWF. Using ex-vivo imaging we could acquire a greater number of MR modalities at higher resolution than would be feasible in-vivo. In addition to the structural and relaxometry data, we also acquired Diffusion Tensor Imaging (DTI) data, allowing direct comparison of the sensitivity and specificity of DTI to relaxometry across the whole brain.

To summarise, the aims of this experiment were: Demonstrate the feasibility of MWF imaging using the mcDESPOT technique in a pre-clinical model.

Observe the effects of cuprizone treatment across the entire encephalon using multiple quantitative MR methods.

Use histological validation to assess their sensitivity to demyelination and inflammation.

Methods

Animal model

All experiments were performed under approval of the local King’s College London ethics committee and the UK Animals (Scientific Procedures) Act 1986, according to the Home Office Project License Number 70/8480 (held by Diana Cash). For a thorough description of the cuprizone model, the reader is referred to Skripuletz et al. (2011). Seven adult male C57BL/6J (Harlan, UK) mice were housed communally and freely fed powdered rodent chow for five weeks, while a further eight were fed powdered chow mixed with 0.2% cuprizone (Bis(cyclohexanone)oxaldihydrazone; Sigma Aldrich, Irvine, UK). All animals were weighed weekly. The five week time-point was chosen as heavy demyelination is expected to be present. They were then killed by transcardiac perfusion with ice-cold heparinized saline followed by 4% buffered paraformaldehyde (PFA). The heads were removed, stored in PFA for 24 h and then rehydrated in Phosphate Buffered Saline preserved with 0.05% sodium azide at 4° for a minimum of 30 days (Cahill et al., 2012).

After image acquisition one control and one cuprizone mouse were excluded from the MRI analysis as the cerebellum had been damaged during sample preparation, preventing a good registration in this region, however the rest of the brain could be used for histology. For quantitative histology, high quality sections could be obtained from all cuprizone animals but only five controls.

MRI acquisition

MR Images were acquired using a 7 Tesla pre-clinical MR system (Agilent Technologies). Samples were immersed in fluorinated liquid to reduce susceptibility artefacts (Galden; Solvay) and loaded four at a time (see Fig. 1) into a 39 mm diameter transmit-receive birdcage coil (Rapid GmbH). Three sets of MR images were acquired; a 3D Fast-Spin Echo for structural analysis, a mcDESPOT protocol, and a DTI protocol.

Figure 1 A schematic of the process used to automatically identify and re-orient each individual subject.

(A) A structural image showing how the subjects were arranged in the scanner. (B) The four largest connected components are labelled as the individual subjects. (C) The center-of-gravity (CoG) of each subject is identified, and a line drawn between the CoG and the origin. This defines the approximate rotation angle required for re-orientation. (D) An individual subject after re-orientation and rigid registration to the atlas.

The 3D FSE image had a matrix size of 256 × 256 × 256 with isotropic 112.5 μm voxels, TE/TR = 40/3,000 ms, an echo-train length of 16, echo-spacing 6.67 ms, readout bandwidth 62.5 kHz, and scan time 3 h 25 min. The DTI scans were acquired using a four shot EPI sequence with a 192 × 128 matrix, 40 slices, 150 × 225 × 500 μm voxel size, TE/TR = 43/4,000 ms, 10 averages and scan time was 3 h 7 min. The diffusion preparation consisted of 30 diffusion directions with b = 2,000 and four b = 0 images, with δ/Δ = 4/16 ms.

The mcDESPOT protocol consisted of a SPoiled Gradient Recalled (SPGR) scan, a balanced Steady-State Free-Precession (bSSFP) scan, and an Actual Flip-angle Imaging (AFI) scan for B1 inhomogeneity correction. The SPGR and bSSFP images both had a 192 × 192 × 192 matrix size with isotropic 150 μm voxels. The SPGR images had TE/TR = 5.148/20 ms, with a read-out bandwidth of 25 kHz, and were acquired at 12 flip-angles (4, 5, 6, 8, 10, 12, 16, 20, 24, 26, 28, 30°), scan time 2 h 28 min, and a strong diffusion spoiling scheme. The bSSFP images were acquired with TE/TR = 3/6 ms, a band-width of 62.5 kHz, also 12 flip-angles (8, 9, 10, 12, 15, 20, 30, 40, 50, 55, 60, 65°) and scan time 2 h 57 min. The extreme flip-angles were chosen to be the optimal for the expected single-component T1 and T2 values (Wood, 2015). The AFI scan had a matrix size of 96 × 96 × 96, isotropic 300 μm voxels, TE/TR1/TR2 = 4.304/20/100 ms, bandwidth 15.625 kHz, a flip-angle of 55° and scan time 37 min (Yarnykh, 2007; Yarnykh, 2010). The acquisition protocol for mcDESPOT allows the calculation of standard relaxometry (T1&T2) maps from the same data.

MRI analysis

The MR Images were first converted to NIFTI format from the manufacturer’s proprietary format, and then were processed using a combination of FSL (Jenkinson et al., 2012), ANTs (Avants et al., 2011) and in-house C++ software utilizing the ITK library, available from https://github.com/spinicist/QUIT. The processing pipeline consisted of several steps. The following operations were carried out in the native space of the acquired images.

A Tukey filter was applied in k-space to remove high frequency noise (Tagge et al., 2016). This is a common reconstruction step on clinical scanners but is not implemented on our system.

The B1 map was calculated from the AFI scan (Yarnykh, 2007). The single-component T1 map was then calculated from the SPGR data using the B1 map to correct the flip-angles. Because we acquired multiple flip-angles (required for mcDESPOT), we used a non-linear Levenberg-Marquadt algorithm instead of the common linearization method to fit the data. An initial value of 1s was chosen for T1 and no issues were observed with convergence to local minima.

T2 and off-resonance maps were calculated from the SSFP data, B1&T1 maps using the DESPOT2-FM method (Deoni, 2009). To improve the speed of the fitting procedure, instead of the original stochastic fitting method we used a bounded non-linear trust-region optimizer (Zhu et al., 1997). To ensure the global optimum was found four different starting points for off-resonance were ranging from −1/(2*TR) to +1/(2*TR) were tried. The starting point for T2 was set to T1/10. Yarnykh & Yuan (2004) lists values of T2/T1 ranging from 0.044 to 0.065 in various in-vivo and ex-vivo samples, but we observed fewer fitting failures using a larger initial value.

The mcDESPOT parameter maps, consisting of the MWF, Intra-Extra cellular Water Fraction (IEWF), Free Water Fraction (FWF), T1&T2 values for each fractional component, and the myelin water residence time (τM), were calculated using a Gaussian-prior stochastic Region Contraction method (GRC) (Deoni & Kolind, 2014). The mcDESPOT model is known to be difficult to fit, particularly close to banding artefacts in bSSFP data (Lankford & Does, 2012; Hurley & Alexander, 2014). In order to stabilize the fit, we applied some additional simple heuristics; the off-resonance value of each component was fixed to the value calculated from DESPOT2-FM and we weighted the residuals of the bSSFP data by 3sin2(ϕ + Ψ)/4, where ϕ is the phase-increment value and Ψ is the accrued phase due to off-resonance in each TR. This weighting scheme ensures that data in the banding artefacts of one phase-increment is ignored in preference to data from the other phase-increments. The fittings ranges for the mcDESPOT parameters were chosen by observing the single-component T1&T2 values in regions of white and grey matter and are given in Table 2.

FSL topup and eddy were used to remove distortion and eddy current artefacts in the raw diffusion data (Andersson, Skare & Ashburner, 2003; Andersson & Sotiropoulos, 2016). We did not acquire data with a reversed phase-encode direction as the manufacturer’s sequence does not have this option. Instead, we synthesized a standard spin-echo image from the T1&T2 maps using the TE/TR of the diffusion sequence. The resulting image had no distortion and diffusion weighting, and was given to topup as a b = 0 image, which could then undistort the acquired diffusion images. The DTI parameter maps were then calculated and consisted of Mean Diffusivity (MD), Axial Diffusivity (AD), Radial Diffusivity (RD) and Fractional Anisotropy (FA) using FSL dtifit.

Table 2 Lower and upper fitting bounds for the mcDESPOT parameters.

The units are milliseconds for all parameters except the MWF and FWF.

Parameter	T1M	T2M	T1IE	T2IE	T1FW	T2FW	τM	MWF (%)	FWF (%)	
Lower	600	6	1,000	40	1,800	120	6	0.1	0	
Upper	1,000	20	1,400	90	5,000	2,500	350	35	100	

The mcDESPOT processing produces ten separate parameter maps (ignoring the B0 and B1 parameter maps that correct for field inhomogeneities). However the MWF, IEWF and FWF are defined as fractions that must sum to one, and so are not independent parameters. Hence, of these only the MWF was used for statistical analysis. Of the remaining parameters the myelin water residence time τM could potentially be an indicator of myelin sheath integrity. However, as will be shown below the current mcDESPOT methodology cannot reliably fit this parameter, so we did not analyse it further.

The following procedure was then used to split the images into individual subjects and register them to a common space: The structural scan was bias-field corrected (Tustison et al., 2010), thresholded and the four largest connected-components were identified. These components were used as masks to separate each subject from the others. The center-of-gravity of each image was then calculated. Because the subjects were scanned in a consistent orientation (with the base of the brain towards the center of the sample tube), we calculated a simple rotation around the tube axis and translation that moved the subjects to the center of the field of view and oriented them in a standard manner. A simple rigid registration was then performed between each subject and an atlas image (Dorr et al., 2008) to ensure all samples were approximately aligned. This process is illustrated in Fig. 1.

A template image was constructed using the 3D FSE and FA images from all subjects in the study (Avants et al., 2010). The resulting 3D FSE template was then non-linearly registered to the atlas image. Including the FA maps in the registration process improved the alignment of the external capsule between groups (see Discussion).

All subjects were non-linearly registered to the study templates using their FSE and FA images. Logarithmic Jacobian determinants were calculated from the inverse warp fields in standard space to estimate apparent volume change. The transforms from native to study template, and from study to standard space were concatenated and applied to all relaxometry and DTI parameter maps to align them to the template. These images were resampled to match the voxel size of the template using a Gaussian interpolator. The full-width half-maximum of the interpolator was set to 100 μm for the relaxometry data and 125 μm for the DTI, due to their differing acquisition voxel sizes.

A brain parenchyma mask was created from the atlas labels by excluding Cerebrospinal Fluid (CSF) regions. The inverse transforms from the atlas to the study template and from the study template to each subject were applied to calculate the brain volume of each subject.

A group analysis was then carried out on all relaxometry maps, DTI maps and the Jacobian deteriminant images with permutation tests and Threshold-Free Cluster Enhancement (TFCE) using FSL randomize (Smith & Nichols, 2009; Winkler et al., 2014). The brain volume estimates were included as a regressor of no interest in the design matrix when analysing the Jacobian determinants, but not for the parameter maps. Animal weights and total brain volumes were compared using a separate two sample t-Test assuming unequal variance.

Histology

After imaging the brains were removed from the skulls and cryoprotected by immersion in 30% sucrose for at least 72 h. A total of 20 μm thick sections were cut in 12 series on a cryostat, collected onto chrome-gelatin coated slides and stored at −20 °C. Immunohistochemistry was performed on two of the series with washes between each step. After rehydrating in Tris buffer (TBS) for 3 × 5 min endogenous peroxidase activity was blocked by applying 1% hydrogen peroxide (H2O2) in TBS for 30 min at room temperature (RT), followed by a non-specific binding block with 10% skimmed milk powder in TBS with 2% Triton-X (TBS-X) for 2 h at RT. Sections were then incubated in primary antibodies for either microglia (rabbit anti-Iba-1, 1:2,000, 019-19741, Alpha Laboratories) or Myelin Basic Protein (rat anti-MBP, 1:1,000, ab7349; Abcam) diluted in 5% skimmed milk powder in TBS-X overnight at 4 °C. Following three washes in TBS-X sections were incubated in either biotinylated goat anti-rabbit or anti-rat antibody diluted in 5% skimmed milk powder in TBS-X (1:1,000; BA-1000 and BA-9400, respectively; Vector Laboratories Ltd.) for 2 h at RT followed by incubation in avidin-horseradish peroxidase complex (Vectastain ABC Elite, PK-6000; Vector Laboratories) for 1 h.

Immunoreactivity was visualized by incubating sections in 0.05% diaminobenzidine and 0.01% H2O2 for up to five minutes with exact timing being determined by the depth of colour of the sections. Sections were then rinsed in TBS and dehydrated in increasing concentrations of industrial methylated spirits (IMS) followed by xylene before cover-slipping with DPX mounting medium (Sigma Aldrich, UK).

A third series was stained with Luxol Fast Blue (LFB) to visualize myelin. Sections were placed in a 50/50 IMS/Histoclear solution overnight to remove fat from the tissue and then hydrated in 95% IMS. Sections were then placed into LFB solution at 56 °C overnight, followed by rinses in 95% IMS and distilled water before differentiating in lithium carbonate for 30 s followed by 30 s in 70% IMS and a rinse in distilled water. Differentiation was then checked microscopically and differentiation steps repeated until GM was clear and WM well defined. Once completed, sections were dehydrated in increasing concentrations of IMS followed by xylene before cover-slipping with DPX mounting medium. For analysis of all three stains we used three sections approximately −1.58, −1.82 and −2.06 mm posterior from Bregma.

Iba1 stained sections were analysed with a Zeiss Axioskop2 MOT microscope and design-based optical fractionator probe in Stereoinvestigator software (v11.03.1, MBF Bioscience). An ROI was drawn over the corpus callosum, cortex above and hippocampus below covering an area of 20 mm2 in each section using a PlanApo 4× objective. Microglia population estimates were obtained using systematic random sampling with a sampling grid of 300 × 300 μm (Gundersen coefficient of error < 0.1) and a counting frame of 50 × 50 μm. Section thickness was manually defined as 15 μm with a dissector height of 14 μm and guard zones of 0.5 μm at the top and bottom of each frame. All microglia falling with the bounds of the counting frame and not touching the exclusion boundaries were counted using a PlanApo 40× objective at the monitor. Cell density was calculated as the ratio of the population estimate to the volume of the ROI, estimated using the Cavalieri principle (West, Slomianka & Gundersen, 1991).

Sections stained for MBP and LFB were analysed using thresholding techniques. The MBP sections were first pseudocoloured in Aperio Imagescope (v12; Aperio Technologies Inc.) where staining in each pixel of the corpus callosum ROI was categorised into one of three levels and a snapshot at ×4 magnification was then analysed by ImageJ (v 1.50b, NIH) software. The analysis included removing of background and measuring the percent coverage of the appropriate peak corresponding to corpus callosum staining. The same method was applied to LFB but omitting the pseudocolouring step and using Otsu threshold parameters in ImageJ. All results were statistically analysed in Prism v6.07 (GraphPad Software Inc.) using unpaired T-tests.

Results

The MR parameter maps and statistic images are available online (DOI: 10.6084/m9.figshare.3495848).

At the end of treatment the mean weight of the control and cuprizone groups were 27.5 ±2.5 g and 21.8 ± 1.2 g, respectively, which were significantly different when assessed with a two-tailed T-test (p = 0.0004). However the mean brain volumes were 387.8 ± 10.1 mm3 and 381.5 ± 9.0 mm3, which was not a significant difference (p = 0.26).

A single slice through the quantitative relaxometry parameter maps for a single control mouse is shown in Fig. 2. WM and GM are clearly distinguishable in the single-component maps. The T2 map better distinguishes the hippocampus and third ventricle at the base of the brain. MWF is clearly visible in the expected regions, with values of approximately 20% in the corpus callosum, slightly higher in the cerebral peduncles and optic tract and less than 5% in GM regions. IEWF is approximately the inverse of the MWF, but a large amount of Free Water is indicated in the third ventricle.

Figure 2 Illustrative relaxometry maps for a control mouse.

Single-component T1&T2 maps are in the left-most column. There are ten mcDESPOT parameter maps—T1&T2 and the fractional amount of three water pools (Myelin, Intra-Extra cellular and Free), and the myelin residence time (τM). WM tracts are clearly visible in the MWF map. The T1&T2 maps of the Free Water (CSF) pool have been omitted.

The myelin T1&T2 maps are fairly flat across the brain, indicating that the fitting routine finds fairly consistent values for these parameters. The exception to this is that the T2 of myelin water in the cerebral peduncles and optic tract appear to be lower than that found in the corpus callosum. The T2 of the IE-water shows some differences between WM and GM, but less than is found in the single-component T2 map. Although the myelin residence time (τM, defined as the mean time a water molecule stays in the myelin pool before exchanging to the IE-pool) shows good contrast, it must be remembered that this parameter is not well defined outside of WM tracts where there is close to 0% MWF (see below).

Similarly, Fig. 3 shows the DTI parameters from a single control mouse. These appear more blurred than the relaxometry maps due to the larger acquisition voxel size, however there is still contrast between WM and GM. AD is visibly higher than RD particularly in the hippocampus. FA shows good contrast between GM and WM, especially in the cerebral peduncles and optic tract.

Figure 3 Single subject maps of the four DTI parameters—Mean, Radial and Axial Diffusivity and Fractional Anisotropy.

These have lower resolution than the relaxometry data but the WM tracts are still visible. MD, AD & RD all show similar contrast, but the absolute value of AD is higher than RD.

Figure 4 shows three axial slices through the group average T1, T2, MWF & DTI maps at the level of the striatum, corpus callosum and arbor vitae of the cerebellum. The control group is presented on the left of each slice and the cuprizone group average on the right. Increases in T1&T2 are obvious in the corpus callosum and cerebellum, with a corresponding decrease in MWF. T2 increases are also visible outside of the WM tracts. Similar effects are present for MD, RD, and AD, while changes in FA are less evident.

Figure 4 The mean control maps compared against cuprizone in three slices.

A control animal (CTRL) is presented on the left of each slice and one treated with cuprizone (CPZ) on the right. Decreased MWF (red arrows) is obvious in WM. Similar changes are visible in other maps except for FA where the corpus callosum is visible in cuprizone animals (green arrows).

Figure 5 overlays the study template with the difference in group means for all parameters, thresholded at FWE-corrected p < 0.05. At this threshold a strong decrease in MWF and corresponding increases in T1&T2 of around 100 & 20 ms, respectively can be detected in the corpus callosum and arbor vitae. The detected regions of T2 change are much larger than for T1, and extend into the cortical and subcortical GM, including the sensorimotor cortex and the dorsal hippocampus. Increases in all diffusivities are also evident in corpus callosum, cerebellum, and thalamus. Changes in FA are restricted to a decrease directly in the splenium of the corpus callosum and an increase directly above, a decrease in the cingulate cortex and isolated decreases below the arbor vitae.

Figure 5 Differences in the group means of the volume change and quantitative parameters overlaid on the study template, thresholded at FWE-corrected p < 0.05.

The areas of significant change differ for each parameter, indicating different sensitivity and specificity to demyelination and inflammation.

Regions of both volume increase and decrease were found in the TBM data. Large increases in volume were found in the splenium of the corpus callosum, external capsule, and inferior parts of the arbor vitae. These correspond well to areas of change indicated in the T1 and MWF maps. Decreases were found in the cortex, striatum, dorsal hippocampus and fimbria, which do not appear to overlap with changes in the quantitative parameters. For reasons of space not all these regions are shown in Fig. 5, but can be viewed in the downloadable results.

Figure 6 shows histology slices for the LFB and MBP stains at approximately the same position as the MRI slices in Figs. 4 and 5, again with a control animal on the left and a cuprizone animal on the right. Decreases in both stains are clearly evident in the cuprizone animal, marked by arrows. In the corpus callosum and external capsule the MBP stain appears to show a decrease towards the edges of the tract, with MBP still present in the center. The cerebellar WM is demyelinated in areas surrounding the cerebellar nuclei.

Figure 6 Representative histological sections stained with LFB and for MBP at approximately the same levels as the MRI.

A control animal (CTRL) is presented on the left of each slice and one treated with cuprizone (CPZ) on the right. Widespread decreases in both stains are marked with arrows in the corpus callosum. Distortion of WM tracts is also evident.

Figure 7 shows equivalent slices through the Iba1 staining for microglial activation. Zoomed areas are marked with boxes, showing the distinct shape of activated microglia in the cuprizone animal in the same areas that show demyelination in the LFB and MBP stains. Microglial activation appears less dense in the cerebellar nuclei compared to the corpus callosum. Figure 8 shows the quantitative histology results in the corpus callosum. Significant decreases in LFB and MBP staining, and increases in the number of Iba1 positive cells were found in the ROIs used. For LFB the mean intensities for control and cuprizone were 73.0 ± 4.3 and 26.8 ± 3.4, respectively (p = 0.0001), for MBP they were 46.6 ± 5.5 and 26.8 ± 3.4 (p = 0.0076) while for Iba1 the population counts were 4,775 ± 420 and 11,963 ± 1,513 (p = 0.0039).

Figure 7 Representative histological sections stained for Iba1.

Activated microglia (zoomed boxes) can be seen in the same regions where LFB and MBP staining indicate demyelination.

Figure 8 Quantitative histology results for the LFB, MBP and Iba1.

ROIs are indicated on the histology images. Significant differences were found for all three stains, confirming the model functioned correctly.

Figure 9 shows a single slice of the Co-efficient of Variation (CoV) for selected parameter maps. The CoV for T1 is excellent, and is less than 5% throughout almost the entire parenchyma, while T2 is marginally worse. The CoV of MWF is highly region-dependent. In GM it is consistently above 10% and approaches 30% in some areas. This is perhaps expected given the low (<5%) absolute value of MWF in these regions. However, even in WM tracts the CoV is generally close to 10% and does not fall below 5%. The CoV of τM shows that this parameter is difficult to fit. Counter-intuitively, in GM areas the CoV appears low while in WM areas it is high. However, it must be remembered that in GM areas there is close to 0% MWF, so here the fitting procedure simply converges to the center of the fitting range. In WM areas, where there should be sufficient MWF to fit a valid τM, the CoV map increases indicating that there is insufficient information to fit this parameter correctly.

Figure 9 A coronal slice through the Co-efficient of Variation maps for selected parameters.

The colourmap for MWF, τM and FA was chosen to emphasise the different color scale. The CoV is smallest for T1&T2, MD, RD & AD and highest for FA & MWF.

For the DTI parameters the diffusivity parameters have a mostly acceptable CoV (<10%) that increases slightly in WM regions. We attribute this to partial volume effects and residual mis-registrations arising from the large voxel size in the anterior-posterior direction for the diffusion acquisition. FA has a high CoV that is above 10% in much of the parenchyma.

Discussion

In this experiment, we aimed to demonstrate the use of mcDESPOT in the cuprizone model, with images acquired over the entire brain, and compare the sensitivity and specificity of multiple quantitative MR methods to demyelination. However, the presence of inflammation in the cuprizone model is a significant potential confound that has not been adequately discussed in previous MR literature.

Validation of mcDESPOT sensitivity to myelination

A major aim of this study was to provide a pre-clinical validation of the MWF as measured by mcDESPOT as sensitive and specific to myelination state. In this regard the study can be regarded as success, principally because in contrast to Thiessen et al. (2013) we found a non-zero MWF in healthy control animals, and then observed a decrease in MWF in cuprizone treated mice.

However, there are some important caveats to this apparent success. The first is that it is not immediately clear what advantages mcDESPOT brings over conventional relaxometry, given the extra acquisition time and extensive processing required to produce the parameter maps. As shown in Fig. 5, the regions of significant change detected in T1 are most constrained to WM, the regions of T2 change extend a long way into the GM, and MWF is somewhere in-between. No regions of significant change were detected in the MWF that were not also detected in either the T1 or T2 map. Due to the co-localisation of inflammation with demyelination in this model, it is difficult to disentangle the impact of these different mechanisms to the MR parameters.

Moreover, because the MWF is defined as a fraction of total water in a voxel, it is obvious that a change in the absolute amount of IE-water will by definition change the MWF, although there has been no change in the absolute amount of myelin water. This means that by definition the MWF can only be sensitive and not specific to myelination state. As currently formulated, due to the need to normalize intensities between the SPGR and bSSFP acquisitions, mcDESPOT cannot be adapted to image absolute myelin and IE-water content, so further work is clearly needed in this area.

The final caveat is that the mcDESPOT model is difficult to fit correctly, and this has been remarked upon elsewhere (Lankford & Does, 2012; Zhang et al., 2014; Bouhrara & Spencer, 2015; Bouhrara et al., 2015). We found a lower MWF than has been reported for in-vivo mcDESPOT human studies, where typical values in WM are over 20% rather than the 10–20% reported here. These lower values are close to those reported by MET2 studies in human in-vivo and ex-vivo studies. However, we found these values to be sensitive to the fitting ranges used, in particular for τM, the residence time of water in myelin. As described above, this parameter is meaningless in GM regions and simply converges to the center of any chosen fitting range.

In WM, where the residence time is well defined, we found values below 50 ms. This is significantly shorter than that reported for human studies. This could be attributed to species differences, or the process of PFA fixation, which is known to disrupt biological membranes and introduce holes into otherwise impermeable structures (Zhang et al., 2012). This will increase the rate at which water can move between the myelin and IE-water pools. Hence, we do not think the low residence times are artefactual, but they do conflict with both the widely used MET2 model and the results of Bouhrara et al. (2015), which do not include exchange because it has been assumed that exchange is slow in relation to T2. Given the high CoV for τM these values should be treated with some caution and further work is needed to improve the accuracy with which this parameter can be extracted from mcDESPOT data.

The cuprizone model

The above results show that the effect of cuprizone treatment is not limited to the splenium of the corpus callosum. To the best of our knowledge this is the first study to use MRI to show extensive cuprizone-induced changes throughout both the cerebrum and cerebellum. In our data the effect of demyelination was particularly striking in the arbor vitae, which contains the deep cerebellar nuclei surrounded by heavily myelinated tracts. Cerebellar demyelination and inflammation has been previously shown in histological examinations, although changes in the cerebellar cortex were less pronounced and delayed (Groebe et al., 2009; Skripuletz et al., 2010). A recent experiment where rats were administered cuprizone and then serially imaged suggested that demyelination started in the cerebellum and progressed forwards in the brain (Oakden et al., 2016).

We found no difference in total brain volume, despite confirming that cuprizone causes significant weight loss (Nystad et al., 2014). We did not test for behavioural changes in our model. It is well known that despite causing profound changes in the brain, cuprizone treatment causes no large behavioural changes and only subtle motor and cognitive deficits, which would require specialized equipment to test for (Skripuletz et al., 2011).

The effect of cuprizone in the principal areas of the arbor vitae, splenium and external capsule is very strong in our data. In the arbor vitae, the change in T1 is approximately 20%, and in T2 it approaches 50%. This means that despite our low group sizes the statistical analysis appears extremely robust, with many voxels meeting the p < 0.05 threshold even using the conservative FWE multiple comparisons correction. T2 but not T1 detects changes in GM regions such as the cortex and thalamus, which are known to also be affected by cuprizone (Gudi et al., 2009; Fjær et al., 2013; Goldberg et al., 2015). Even more widespread changes in T2 could be seen using FDR instead of FWE correction (available in the on-line dataset), whereas changes in T1 remain restricted to areas that are heavily myelinated in healthy animals.

We also detected changes in several quantitative parameters in the hippocampus, where myelination and inflammatory changes have been well characterised by histology (Skripuletz et al., 2011; Goldberg et al., 2015). This is again a region which most imaging studies have neglected to either analyze or report. Our results indicate the possibility that previous imaging studies to document the longitudinal profile of the cuprizone model have been missing important areas of the brain, and inclusion of these areas in future studies is necessary.

It was beyond the scope of this paper to perform a fully quantitative histological evaluation of all brain areas in which the MRI changes were detected. Many studies to date have shown extensive histological and immunohistochemical changes due to cuprizone treatment, which match both our qualitative (Figs. 6 and 7) and quantitative results (Fig. 8). We limited our myelination quantification with LFB and MBP to the key areas of rostral corpus callosum, and microglial assessment with Iba1 to the surrounding cortex and hippocampus. As expected, these showed a robust and significant decrease in myelination in cuprizone mice, and a profound increase in number of activated microglia. It is also clear from qualitative observations of histological slides that areas of demyelination are spread throughout the brain and particularly salient at the external capsule, hippocampus and arbor vitae. The affected WM areas are accompanied and surrounded by clearly increased Iba1 staining and visibly enlarged microglial cell bodies, indicating ongoing microglial hypertrophy and hyperplasia. The colocation of demyelination and inflammation in the cuprizone model means that it is difficult to rule out a contribution from inflammation to changes in the MR parameters.

We believe this study is the first to estimate brain volumes and apparent volume change in the cuprizone model. Localized volume changes (both increases and decreases) were widespread and extended to regions that showed no change in relaxometry or DTI parameters. Volume increases were located in the corpus callosum, dorsal hippocampus, arbor vitae and sensorimotor cortex. These were corroborated by histology as apparent enlargement and distortion of WM tracts, reflecting previously reported underlying axonal swelling, damage and cellular infiltration (Song et al., 2005; Xie et al., 2010). In addition, we also found bilateral decreases in the frontal, cingulate and retrosplenial cortices as well as the caudate putamen and several other subcortical GM areas.

The lack of change in FA (discussed further below) was beneficial for this study as this parameter could be used to improve the registration process. Initially, we followed standard practice and used only the 3D FSE image for template creation and registration. However, due to the strong effect of cuprizone the FSE contrast in several WM regions is reversed compared to control animals. This led to subtle misalignments between the two groups, principally of the external capsule, resulting in nonsensical changes in volume and FA outside WM tracts. Incorporating the FA maps into the registration correctly aligned the external capsule in the cuprizone group.

This demonstrates a difficulty in using current automated registration methods, as fundamentally they assume that all input images are equivalent and this is clearly not the case when gross pathology or even subtle changes in T1&T2 are present (Cousins et al., 2013). Although non-linear deformation algorithms include regularisation methods to prevent excessive volume changes (van Eede et al., 2013), this was not sufficient in our data to prevent mis-registrations. However, incorporating a parameter that did not demonstrate gross changes between groups (FA) stabilised the registration and yielded a high quality result.

Diffusion measurements in the cuprizone model

DTI has become an extremely widespread method of assessing WM health in clinical studies (Jones, Knösche & Turner, 2013), but interpretation of the diffusivity and FA parameters can be difficult. In particular, RD and FA are often assumed to be a marker of myelin “integrity” (Song et al., 2005; Wheeler-Kingshott & Cercignani, 2009; Janve et al., 2013). Our results indicate two issues; firstly that both RD and AD (and hence MD) are sensitive to demyelination, and that FA appears to be far less sensitive to changes in myelination than diffusivities.

There are conflicting literature results for diffusion changes in the acute phase of the cuprizone model, which are summarised in Table 3. Results from the chronic (greater than six weeks) phase of cuprizone treatment have been omitted, as these do not relate to our time point. However, the reported results also vary widely in diffusion protocols, importantly the number of diffusion directions and strengths. It is possible that earlier papers using only six directions could not correctly calculate the tensor correctly for tracts in all orientations. Some caution should be applied to interpretations comparing diffusion imaging from in-vivo and ex-vivo subjects, because as discussed by Zhang et al. (2012), diffusivity parameters may alter as a result of tissue fixation. Another contributory factor is different lengths of cuprizone treatment, as the levels of demyelination and inflammation in cuprizone undergo progressive change between four and six weeks of treatment (Skripuletz et al., 2011). In particular, at week four demyelination is incomplete whereas microglial activation is highest, but by week five demyelination is complete but microglial activation is decreasing. Hence, as discussed for relaxometry above, there are competing effects of inflammation and demyelination on diffusivity measurements which are hard to disentangle using this model. Recent work (Guglielmetti et al., 2016) supports this hypothesis, as they found an initial decrease of diffusivity after three weeks of cuprizone treatment, followed by an increase at six weeks.

Table 3 A summary of DTI protocols and findings in the acute phase of cuprizone treatment.

Results past six weeks are omitted for brevity. The findings vary widely, but this is at least in part due to the different imaging protocols and treatment lengths.

Citation	N	δ/Δ (ms)	b (s/mm2)	In-/Ex-vivo	Results	
Song et al. (2005)	6	4/11	1,600	In	6w:RD↑	
Sun et al. (2006)	6	10/25	768	In	4w:AD↓, 6w:AD↓RD↑	
Wu et al. (2008)	2*	*	700	In	4w:AD↓RD↑	
Xie et al. (2010)	6	8/25	768	In	4w:AD↓	
Zhang et al. (2012)	6	3/15	1,000	In	4w:AD↓ 6w:RD↑	
			1,700	Ex	4w:FA↓RD↑, 6w:FA↓RD↑	
Chandran et al. (2012)	30	3/10	750	In	5w:FA↓	
Thiessen et al. (2013)	7	6/14	1,000	Ex	6w:FA↓AD, RD↑	
Guglielmetti et al. (2016)	30	5/12	400–2,800	In	3w:MD, AD, RD↓, 6w:MD, AD, RD↑	
Jelescu et al. (2016)	30	2/16	1,000/2,000	In	6w:RD↑	
Notes:

N: Number of diffusion encoding directions.

* Wu et al. (2008) did not report their δ/Δ values, and used two directions oriented parallel and perpendicular to the corpus callosum instead of calculating the tensor.

The limited areas of change in FA detected in the cuprizone model were surprising. There is only one area of convincing change, a decrease directly at the splenium of the corpus callosum, with an increase directly above. Comparison to histology suggests that this region of the corpus callosum appears to distort, reducing the ordered nature of the axons. Smaller potential changes are located in the cingulate cortex and on the edges of the arbor vitae, but there are no widespread effects throughout WM tracts as for the other parameter maps. This suggests that FA is insensitive to the myelination state of WM, although this may be due to the increased variance in the FA measurements.

It was beyond the scope of this work to include an assessment of axonal damage via neurofilament histological staining. However recent work indicates that during the acute phase (less than six weeks) of cuprizone treatment axonal damage should be minimal (Schregel et al., 2012; Alme et al., 2015), and only starts to accumulate during the chronic phase (Lindner et al., 2009). This suggests that the changes we see in FA are due to changes in the extracellular matrix or organisation of axons, rather than outright damage.

Comparison of DTI and relaxometry data

Our results are broadly in line with those of Santis et al. (2014), who compared mcDESPOT and DTI in healthy human subjects. They found a low correlation between FA and measures of myelination. They also found a similar trend of parameter variance, with T1 having the best performance and FA & MWF the worst. They found that T2 measurements performed significantly worse than diffusivities, while we found it performed slightly better. We attribute this to the increased number of phase increments in our bSSFP acquisition, which we found markedly decreased banding artefacts.

We calculated Pearson’s correlation ratio between the quantitative parameters across the entire brain and all mice using AFNI (Cox, 2012). Slices through the splenium and cerebellum are shown above and below the diagonal, respectively in Fig. 10. Red colors indicate strong positive correlations, and blue indicates strong negative correlations. Perhaps unsurprisingly, the relaxometry parameters and diffusivities form two separate groups of highly correlated parameters.

Figure 10 Pearson’s Correlation Ratio between each pair of MR parameters.

A slice through the external capsule is shown above the diagonal and a slice through the cerebellum below the diagonal. Correlations between parameters are very strong (but both positive and negative) in the arbor vitae and splenium.

T1&T2 are highly correlated in WM regions, and particularly so where cuprizone acts, but are less strongly correlated in GM. They are both negatively correlated to MWF in similar patterns. Correlations between diffusivities and T1&T2 are poor except in areas where cuprizone has a strong effect. Correlations between FA and the other parameters are poor. MD, AD and RD are highly correlated in both GM and WM, except for AD and RD in the cerebral peduncles, fimbria and optic tract. These WM tracts are known to be resistant to cuprizone damage at five weeks (Goldberg et al., 2015), and we observed no qualitative changes in our histology in these regions. Hence, the low correlations are likely a result of different WM organisation in these regions compared to the highly ordered corpus callosum.

This analysis underscores the severity of the cuprizone model at five weeks—in the primarily affected regions, the difference between model and control is so complete that correlations across the entire dataset approach one, and we expect the same would hold true for a histological analysis. It would be useful to better understand the contribution of myelin and inflammation to the different MR parameters, similarly to how Desmond et al. (2016) developed a model of the contribution of myelin and metals to T1&T2. In our opinion, due to the strength of the correlations the single time point presented here is insufficient to build such a model. However, microglial activation is known to precede demyelination in the cuprizone model by approximately a week (Skripuletz et al., 2011), and is required for demyelination to occur (Praet et al., 2015). Hence, a longitudinal study with several earlier time points should observe a differential build up of inflammation and demyelination, allowing the two effects to be separated.

Conclusions

This experiment demonstrates that T1&T2, the MWF, and diffusivities are sensitive to demyelination but not necessarily specific, due to confounding co-localized inflammation in the cuprizone model. FA appears insensitive to myelination state. In addition, we found that cuprizone causes localized volume changes in the mouse brain. Collectively these results show that whole brain acquisition and analysis is crucial to full understanding of the cuprizone model. We propose that similar methods would be beneficial when using MRI to study other preclinical models of neurodegeneration to better understand and refine the knowledge of brain pathology.

We acknowledge many useful discussions with Jonathan O’Muircheartaigh. We thank Professor Federico Turkheimer for reading the paper and making helpful suggestions.

Additional Information and Declarations

Competing Interests

Author Contributions

Animal Ethics

Data Deposition

The authors declare that they have no competing interests. Samuel A. Hurley is an employee of Synaptive Medical, Toronto, ON, Canada.

Tobias C. Wood conceived and designed the experiments, performed the experiments, analyzed the data, contributed reagents/materials/analysis tools, wrote the paper, prepared figures and/or tables, reviewed drafts of the paper.

Camilla Simmons performed the experiments, analyzed the data, wrote the paper, prepared figures and/or tables, reviewed drafts of the paper.

Samuel A. Hurley contributed reagents/materials/analysis tools, reviewed drafts of the paper, provided the DESPOT sequences for the Agilent scanner.

Anthony C. Vernon conceived and designed the experiments, reviewed drafts of the paper.

Joel Torres analyzed the data.

Flavio Dell’Acqua reviewed drafts of the paper, assisted with the diffusion sequence for the Agilent scanner.

Steve C.R. Williams conceived and designed the experiments, contributed reagents/materials/analysis tools, reviewed drafts of the paper.

Diana Cash conceived and designed the experiments, performed the experiments, analyzed the data, wrote the paper, prepared figures and/or tables, reviewed drafts of the paper.

The following information was supplied relating to ethical approvals (i.e., approving body and any reference numbers):

All experiments were performed under approval of the local KCL ethics committee and the UK Animals (Scientific Procedures) Act 1986, according to the Home Office Project license no 70/8480 (held by Diana Cash).

The following information was supplied regarding data availability: Code–http://github.com/spinicist/QUIT;

Quantitative maps & stats images–https://figshare.com/articles/Whole-Brain_Ex-Vivo_Quantitative_MRI_of_the_Cuprizone_Mouse/3495848.

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
