# Peer review of "Whole-brain ex-vivo quantitative MRI of the cuprizone mouse model"

_PeerJ, doi:10.7717/peerj.2632_

## Round 0.1 · original submission · Major Revisions

Dear Authors,

Two peer reviewers have mentioned very important comments that your team should take into serious consideration.Please do the necessary revisions and resubmit to PeerJ for re-review when the revisions are completed.

·

Basic reporting

"No Comments"

Experimental design

"No Comments"

Validity of the findings

"No Comments"

Additional comments

1. cuprizone is demylination and remylination model. You just limited your study to validate demyleination process. It would be interested to study both processe and suggest a validation.

2. I suggest you compare your cuprizone model with other MS model.

3. Your diffusion weighted imaging is so limited and you did not compare it with other imaging approaches.

4. Your sample number is low

Reviewer 2 ·

Basic reporting

This article presents an in-depth MRI study using T1/T2, myelin water fraction and DTI parameters to study demyelination in cuprizone mouse model and supported with myelin and microglia histology.
This paper attempt to address hotly debated topics in MRI, and I would like to see this manuscript published.
The manuscript is in a very good shape. It is very well written and the figures are clear.

Experimental design

This paper aims to test the sensitivity of myelin water fraction imaging using mcDESPOT in cuprizone demyelination model, and compare the results with standard quantitative MR methods.

The experiment is well laid out, but I have a few comments:
Line 101, it is not clear if this total time include DTI. Please add the acquisition time for each modalities, and also for mcDESPOT acquisitions.
What is the b values and the d/D for DTI?
130: What is L-BFGS-B?
151; I don’t understand this sentence “Instead, we synthesized an image from the T1&T2 maps and the TE/TR of the diffusion sequence.” What image?

Validity of the findings

The data presented is robust, and the control is present.

Comments:
Line 238: is there any behavioural difference between control and cuprizone mice?
Line 274: Perhaps should add a comment about large areas of changes shown by T2?

Figure 6. Do you still have the samples to do neurofilament axonal staining? Perhaps FA is sensitive to the arrangement of axonal fibre, so if they are intact, then this would be expected.
FA has been used to study embryonic brain development, where they are still completely unmyelinated.

319: Perhaps you use low/mid b-value ~750 and high b-value ~3000 s/mm2 then you can distinguish between inflammation and demyelination components?

414: Something is missing “decreases in FA after . ”

Lines 410-417 and 419-425 are rather confusing. Perhaps it warrants a table to present/summarize it clearly.
Can you add treatment length, the results of microglia, myelin and axonal staining (if available) in this table? Also add their b-values and directions, in vivo/ex vivo.

446: Performance comparison between DTI and relaxometry data. I think you will need to do regression analyses between the MRI parameters and the myelin and microglia histology staining (and axonal staining?) to test for their correlation strengths.
- What is the effect of microglia infiltration and/or demyelination to the MRI data?
- Are there some areas which has demyelination but not so much increase in microglia activation? Do these areas have different MRI parameters to areas with both strong demyelination and microglia activation?

---

## Round 0.2 · accepted · Accept

Dear Authors,

Thank you for your revised manuscript which has been accepted for publication in Peer J.Please do some minor editing at production, as noted by Reviewer 2

·

Basic reporting

"No Comments"

Experimental design

"No Comments"

Validity of the findings

"No Comments"

Reviewer 2 ·

Basic reporting

This report was written in clear english and presented clearly.
The authors have addressed my previous queries and comments satisfactory.
I recommend this manuscript for publication in PeerJ.

Experimental design

This paper contributes broadly to MRI research in the areas of white matter pathology and MS model. It has especially tease out the correlation and sensitivity between T1/T2 and DTI parameters to measure changes in WM content.

Additional experimental details have been added as requested.

Validity of the findings

Additional results and discussion have been added to make this paper clearer.

Additional comments

Minor editing:
line 118: delete "also"
Fig 6: delete "Again"
line 520: "known"

---

## Author Rebuttal · Round 0.2

We thank the reviewers for their time and input.

There are some minor changes to the figures. The colour scheme and limits in figure 5 was tweaked to emphasise the size of changes in T1 and T2. The layout of figure 7 was changed to accommodate requests from the editorial office. A new figure (10) has been added in order to respond to reviewer 2's comments fully.

Responses to individual points raised by the reviewers follows below.

*Reviewer 1*

*1. cuprizone is demylination and remylination model. You just limited your study to validate demyleination process. It would be interested to study both processe and suggest a validation.*

> The re-myelination process after withdrawal of cuprizone has already been studied in multiple papers, which we have cited (e.g. Skripuletz et al. Histology and Histopathology 2011). Adding an additional time-point in either the de- or re-myelination phase would require running an additional cohort of animals and an entirely new experiment which was beyond the scope of our current work.

*2. I suggest you compare your cuprizone model with other MS model.*

> We used cuprizone as a tool to investigate the link between myelination and quantitative MR parameters. It was not our intention to test its construct or face validity as a model of multiple sclerosis, or to make statements about the utility of mcDESPOT and/or DTI for imaging demyelination in MS patients. Comparing to an additional model is therefore, again, whilst ultimately useful as a future study, beyond the scope of this paper.

*3. Your diffusion weighted imaging is so limited and you did not compare it with other imaging approaches.*

> In the absence of a more detailed comment, it is hard to interpret precisely what the reviewer means by "limited" in this context. We note that comparisons between different diffusion imaging techniques in mice exposed to cuprizone have already been published in several recent papers, which we cite in our work (for example: Jelescu et al. NeuroImage 2016, Guglielmetti et al. NeuroImage 2016, Falangola et al. NMR in Biomed 2014).

> The DTI protocol used in the current study is comparable to standard clinical acquisitions and therefore has translational relevance to such studies. We focused on achieving full brain coverage and high spatial resolution in the time available instead of the multiple diffusion shell acquisitions required for more advanced diffusion methods. This would be interesting for a follow up study, but again is beyond the scope of the current work.

> Moreover, we draw the reviewer's attention to the fact that we did compare the diffusion data to our additional relaxometry (T1,T2 and MWF) measurements. The discussion section on this comparison has been expanded to emphasise

this and in response to reviewer 2's comments.

*4. Your sample number is low*

The demyelinating effect of Cuprizone is so strong (e.g. approximately a 20% change in T1 and close to a 50% change in T2 in the arbor vitae) at the 5 week time-point that group comparisons are still robust despite our low group numbers. However, we cannot exclude the possibility that we were under-powered to robustly detect more subtle demyelination in grey matter. Notably, such changes were apparent when the data were corrected for multiple comparisons using the False Discovery Rate (FDR) at a threshold of $q<0.05$, but not at the more conservative Family Wise Error correction we used. For completeness, we included these new FDR results in the on-line dataset and we have added new discussion of this issue at lines 373-380.

In addition reviewer 1 returned a heavily annotated manuscript. We have responded to as many points as is feasible in this space. Several of the comments were of the form "revise this sentence" without indicating what the reviewer found problematic. While we have endeavored to clarify sentences that were possibly confusing, in the absence of further guidance it is not always possible to respond to such comments.

*Please consider changing the title of the article*

As the reviewer has not specified why they think the title is inappropriate, we have simply changed "Cuprizone Mouse" to "Cuprizone Mouse Model" in the title.

*Reference formatting*

The reference formatting in the review copy have been updated to the Journal's style.

*You should replace throughout of your article to the cuprizone mouse model*

We have changed instances "cuprizone mouse" to "cuprizone model".

*Provide more description of such a unique model*

A sentence has been added giving an overview of the model and related references.

*What are the reasons for limited brain coverage???*

It would be speculation for us to comment why previous studies only had limited brain coverage as this can be attributed to a myriad of reasons e.g. RF coil architecture, the imaging sequence, scanner cost and availability.

*For in-vivo experiment or as ex-vivo experiment*

Both.

*Could you consider changing T1, T2 relaxation time after immersed the tissues in such solution. Also the diffusion coefficient in-vivo experiment are completely differ to ex-vivo samples*

> As stated in the methods section, the samples were rehydrated in PBS for a minimum of 30 days to allow T1 & T2 to stabilize. We have tested this in a subsequent dataset (currently unpublished) and the data shows that T2 increases initially and then plateaus at 30 days, confirming that 30 days is an appropriate duration for these parameters to stabilize. (Shepherd et al. Magnetic Resonance in Medicine 2011).

> We noted in the discussion section that diffusion parameters change between in-vivo and ex-vivo, which may complicate comparisons to in-vivo data (Zhang et al Magnetic Resonance in Medicine 2012).

*You did not describe the course of such treatment*

> We described the course of cuprizone treatment  used in the methods section

*I think you should provide more information about your animal model*

> The reviewer does not state what information they think is missing. The animal model is fully described (mouse strain, age, housing conditions, and how the cuprizone was fed to the animals). We have added a sentence directing readers to a comprehensive review paper.

*How did you decide the demyelination had occurred*

> The time-course of demyelination in the cuprizone model has been well established in other papers (e.g. Skripuletz et al Histology and Histopathology 2011). A five week timepoint was chosen as this is the earliest point where heavy demyelination is expected. We confirmed demyelination had occurred with quantitative histology in the corpus callosum.

*What are the clinical symptoms of each mouse*

> The only obvious symptom of cuprizone treatment is weight loss, there are no other gross neurological defects (Skripuletz et al Histology and Histopathology 2011). As stated in the methods and results, we weighed the mice weekly and observed a large difference between control and treatment groups at the 5 week timepoint.

*How does it (the fluorinated liquid) affect T1, T2 relaxation time measurement and diffusion weighted measurement?*

> The fluid is a perfluoropolyether. It does not contain protons, and thus does not contribute to the measured MR signal. Also, it does not enter the tissue and so does not affect these parameters directly. It improves image quality by reducing susceptibility artefacts at the edge of the head and hence improves the

quantification.

*Not sure what do mean*

We have adjusted the text to refer readers to figure 1, which should clarify how the samples were positioned in the scanner.

*What is the b-value for the diffusion weighted images?*

The b-value has been added to the text.

*Why did you not acquire isotropic resolution???? It is ex-vivo experiment, why you did not run your experiment for long time to acquire higher resolution????*

We mistakenly only included the scan time for the 3D FSE scan in the manuscript. In addition to the FSE scan (3 hours 25 minutes), we acquired mcDESPOT (6 hours total) and the DTI which took 3 hours by itself (with 10 averages to achieve adequate SNR). The complete protocol was a long as possible (12.5 hours) within the timetabling constraints for this experiment. Hence achieving isotropic resolution for the DTI was not feasible in this study.

The scan times and averaging for the DTI have been added to the text. We note that we are not aware of any papers, ex-vivo or in-vivo, that achieved isotropic resolution for DTI imaging in the cuprizone mouse model.

*What are the reasons for the selection of the variable flip-angle?*

The mcDESPOT method requires multiple flip-angles in order to fit the non-linear signal model (Deoni et al Magnetic Resonance in Medicine 2008).

*what is this abbreviation stand for*

Limited memory Broyden-Fletcher-Goldfarb-Shanno with simple Bounds. Given the space that would be required to write this out fully, we have elected to remove the specific name but retain the citation to the paper describing it.

*What is the value for mouse brain*

This information was not available before we conducted our study. Looking at our data, it is approximately 0.04 in white matter but increases in and around the ventricles and non-parenchyma tissue towards 0.1, which explains why the higher starting value was required.

*How did you calculate DTI maps*

We used dtifit in FSL. This has been added to the text.

*I think you should add reference here*

The reference is already in the previous paragraph, as the same atlas is used

in both steps. The text has been clarified to hopefully make this clearer.

*At the end of treatment, I would prefer to replace to the following: After given the cuprizone for five weeks, describe the clinical symptoms of the mice..........*

*What are other clinical symptoms of each mouse such as paralysis and so on*

As already discussed, the mice have no clinical symptoms other than weight loss.

*How did you quantify these values?*

By inspecting the parameter maps using a viewing program. We have made the dataset available so that others can do the same.

(We thank the reviewer for drawing our attention to the inconsistent use of lower/upper case m for the myelin residence time. We have opted to use an uppercase M consistently.)

*FA should provide much better contrast than you presented in your study*

We have reduced the maximum value in the color map for the FA images.

*Axial or radial*

All 3 (axial, radial, mean).  This has been made explicit in the text.

*This can be minimized as you are running ex-vivo experiment*

It was minimized as far as possible in the time constraints of the experiment.

*It is not clear what meant to deliver*

We do not understand this comment.

*What are the values of MWF for control vs. cuprizone mice?*

The reviewer is referred to the second paragraph of the results, where the values of MWF in control mice are stated, and figure 4 where the values in healthy and cuprizone mice are compared graphically.

*Can you present evidence for that?*

Figures 6 and 7 are the evidence.

*Could you present images for non damaged tissues or otherwise zoomed in as you did for figure 7*

The non-damaged (healthy control) tissues are presented on the left of each

slide and cuprizone on the right. The figure caption has been amended to make this clear.

The zoomed boxes on figure 7 were used to show the shape of the activated microglia, in order to confirm that the staining was not due to debris. The stains in figure 6 (LFB & MBP) do not require this confirmation.

*It is well known demyelination model how does correlate with quantification parametric maps*

We refer the reviewer to their earlier comment about deciding how demyelination occurred. The purpose of the quantitative histology was to confirm that the model had functioned as expected.

Correlations are discussed further in response to reviewer 2.

*What are the reasons for not observing the FA changes??*

This was discussed at the end of the section in which this comment occurs.

*What did you use for your experiment??*

30 diffusion directions were used. This was stated in the methods.

*???????*

Our results do not show FA changes in regions that are clearly demyelinated, principally the arbor vitae. There is a paragraph dedicated to this issue in the discussion.

*Reviewer 2 (Anonymous)*
*Basic reporting*
*This article presents an in-depth MRI study using T1/T2, myelin water fraction and DTI parameters to study demyelination in cuprizone mouse model and supported with myelin and microglia histology.This paper attempt to address hotly debated topics in MRI, and I would like to see this manuscript published. The manuscript is in a very good shape. It is very well written and the figures are clear.*

*Experimental design*
*This paper aims to test the sensitivity of myelin water fraction imaging using mcDESPOT in cuprizone demyelination model, and compare the results with standard quantitative MR methods.*

*The experiment is well laid out, but I have a few comments:*
*Line 101, it is not clear if this total time include DTI. Please add the acquisition time for each modalities, and also for mcDESPOT acquisitions.*

Acquisition times have been added for all scans.

*What is the b values and the d/D for DTI?*

These values have been added to the text.

*130: What is L-BFGS-B?*

L-BFGS-B is a bounded non-linear trust-region local optimization algorithm. The text has been amended to make this clearer. There is a lack of good, open-source, bounded optimizers implemented in C++. L-BFGS-B is available in two different libraries and produced good results.

*151; I don't understand this sentence "Instead, we synthesized an image from the T1&T2 maps and the TE/TR of the diffusion sequence." What image?*

We agree the sentence was unclear, and it has been reworded as follows:

"Instead, we synthesized a standard spin-echo image from the T1&T2 maps using the TE/TR of the diffusion sequence. The resulting image had no distortion or diffusion weighting, and was given to topup as a $b = 0$ image, which could then undistort the acquired diffusion images."

This idea was inspired by a posting on the FSL mailing list, where the author of topup (Jesper Andersson) suggested using a non-diffusion weighted spin-echo acquisition as a reference for topup.

*Validity of the findings*
*The data presented is robust, and the control is present.*

*Comments:*
*Line 238: is there any behavioural difference between control and cuprizone mice?*

Cuprizone is known to cause some motor and cognitive deficits, but does not produce large behavioural changes (Skripuletz et al Histology and Histopathology 2011). Discussion of this has been added at lines 364-367.

*Line 274: Perhaps should add a comment about large areas of changes shown by T2?*

We have emphasised that changes in T2 cover a larger area in the results section at line 271 and added some discussion at lines 368-372.

*Figure 6. Do you still have the samples to do neurofilament axonal staining? Perhaps FA is sensitive to the arrangement of axonal fibre, so if they are intact, then this would be expected. FA has been used to study embryonic brain development, where they are still completely unmyelinated.*

Whilst this is an excellent suggestion, unfortunately, we do not have enough sections remaining for neurofilament staining. However, several other papers have previously conducted a similar analysis. An additional paragraph discussing these has been added at lines 452-456. To summarise these

studies, axonal damage is *not* expected after 5 weeks of cuprizone treatment and thus unlikely to be the source of changes in FA.

*319: Perhaps you use low/mid b-value ~750 and high b-value ~3000 s/mm2 then you can distinguish between inflammation and demyelination components?*

This is an interesting suggestion, but beyond the scope of this paper. There is already some work in this area, e.g. Guglielmetti et al NeuroImage 2016.

*414: Something is missing "decreases in FA after . "*

The words "5 weeks" were missing. However, this paragraph has been reworded due to the next comment.

*Lines 410-417 and 419-425 are rather confusing. Perhaps it warrants a table to present/summarize it clearly.*
*Can you add treatment length, the results of microglia, myelin and axonal staining (if available) in this table? Also add their b-values and directions, in vivo/ex vivo.*

We agree that the summary of DTI results could be hard to digest. We have tabulated the DTI protocols, treatment lengths and results as suggested. It was not possible to summarise histology results due to the wide variety of stains and quantification methods. We hope the resulting discussion is now much clearer.

*446: Performance comparison between DTI and relaxometry data. I think you will need to do regression analyses between the MRI parameters and the myelin and microglia histology staining (and axonal staining?) to test for their correlation strengths.*
*- What is the effect of microglia infiltration and/or demyelination to the MRI data?*
*- Are there some areas which has demyelination but not so much increase in microglia activation? Do these areas have different MRI parameters to areas with both strong demyelination and microglia activation?*

We agree that ultimately what is required is a regression between histology and MRI to tease apart the differential impact of inflammation and demyelination. However we are hampered in this by the strength of the cuprizone model at the selected five-week time point. To illustrate this we created an additional figure displaying the correlations between MR parameters. The effect of cuprizone is so strong that in the principal areas of MRI signal changes (e.g. arbor vitae, splenium, external capsule) the correlations are close to unity (i.e. 1).

Interestingly, prior work suggests that the time-courses of demyelination and inflammation are different in the cuprizone model (Skripuletz et al Histology and Histopathology 2011. Hence, it would be more informative to perform regression analysis on a new dataset covering additional time-points with both MRI and histology. The different time-points would then provide the differential response required to separate the effects of myelination and inflammation. A discussion of these issues has been added at lines 464-487.